# Association between Sleep Timing, Being Overweight and Meal and Snack Consumption in Children and Adolescents in Southern Brazil

**DOI:** 10.3390/ijerph20186791

**Published:** 2023-09-21

**Authors:** Denise Miguel Teixeira Roberto, Luciana Jeremias Pereira, Francilene Gracieli Kunradi Vieira, Patricia Faria Di Pietro, Maria Alice Altenburg de Assis, Patrícia de Fragas Hinnig

**Affiliations:** Post-Graduation Program in Nutrition, Health Sciences Center, Federal University of Santa Catarina, Florianópolis 88040-900, Brazil

**Keywords:** midpoint of sleep, eating events, meals, obesity, schoolchildren, bedtime

## Abstract

Sleep timing is one of the dimensions of sleep that refers to the time of day when sleep occurs. It has been included in sleep-related research because of the potential associations between being overweight and the consumption of meals and snacks. This cross-sectional study aimed to investigate associations between sleep timing, meal and snack consumption and weight status in 1333 schoolchildren aged 7–14 years. The midpoint of sleep was used as a sleep timing measure obtained by the midpoint between bedtime and wake-up time, classified as Early, Intermediate, and Late. Schoolchildren in the Early group were less likely to be overweight (OR: 0.83, 95% CI 0.69; 0.99), and had higher odds of mid-morning snack consumption (OR: 1.95, 95% CI 1.56; 2.44) and lower probability to consume an evening snack (OR: 0.75, 95% CI 0.59; 0.94) compared with the Intermediate group. The Late group had lower odds of mid-morning snack consumption (OR: 0.67, 95% CI 0.55, 0.80) than the Intermediate group. The consumption of mid-morning and evening snacks was associated with the Early and Late midpoints of sleep. These results suggest that bedtime and wake-up time are relevant to consuming meals and snacks and may also be related to a greater probability of being overweight in children and adolescents.

## 1. Introduction

In the past few decades, studies on chronotype and sleep have increased due to their important relationship with health [1]. Chronotype is a biological construct that allows for identifying individual preferences in the timing of sleep and wakefulness, and is also classified according to the tendency of individuals to be morning/early or evening/late types [2,3,4]. This construct is directly related to sleep timing, which refers to the time of day that sleep occurs [5] and is commonly measured by bedtime, wake-up time, and the midpoint of sleep. The midpoint of sleep on free days (MSF) is defined as the midpoint between bedtime and wake-up time on free days, and identifies the chronotype [2].

The research on sleep, nutrition and obesity has predominantly focused on sleep duration [6,7]. However, other dimensions of sleep seem to influence health, such as sleep timing, which may be a better predictor of obesity than sleep duration alone [8]. The evidence suggests that later sleep timing may be associated with poorer health outcomes in children and adolescents, such as sleep duration/quality, eating behaviors and physical activity and sedentary behaviors [9]. Delays and shifts in sleep timing appear to cause weight gain due to behavioral and physiological changes [10]. These sleep delays can lead to circadian misalignment, contributing to weight gain, obesity and adverse metabolic health [11]. The circadian system is composed of a central clock (the suprachiasmatic nucleus of the hypothalamus) and peripheral clocks in almost all tissues of the body and imposes a rhythmic control over virtually all our bodily functions; for example, from when we feel drowsy to when we feel hungry [12]. 

Observational studies with children showed associations between later bedtimes, higher energy intake after dinner [13] and increased obesity risk [7,14]. Regarding the midpoint of sleep, a longitudinal study found a positive association between the midpoint of sleep and Fat Mass Index in adolescents aged 12–15 [15].

Studies have shown that morning habits are associated with healthy eating habits, such as breakfast daily [16], compared to children and adolescents with later habits. Meanwhile, late habits are associated with higher unhealthy eating habits [16,17], skipping breakfast, eating higher-energy-dense foods [14,18], lower consumption frequency of fruit and vegetables [18,19] and with being overweight or obese, independent of sleep duration [10,18,20].

However, little is known about the association between sleep timing, the frequency of meals and snacks and weight status in children and adolescents. We hypothesized that children and adolescents with a later midpoint of sleep are more likely to be overweight or obese and consume evening snacks late at night. Moreover, they tend to skip earlier meals, such as breakfast and mid-morning snacks. On the other hand, those in the earlier midpoint of sleep are inversely associated with being overweight or obese and more frequently consume breakfast and mid-morning snacks, in addition to skipping evening snacks. This study aimed to assess the association of the midpoint of sleep with weight status and meal and snack consumption in children and adolescents aged 7–14.

## 2. Materials and Methods

### 2.1. Study Design and Sample

This cross-sectional study was conducted as a part of a school-based surveillance named the Study on the Prevalence of Obesity in Children and Adolescents in Florianópolis, southern Brazil (EPOCA survey), which was enrolled in public and private schools in the urban area. This study was carried out between November 2018 and December 2019. The EPOCA survey aimed to investigate the prevalence of obesity and its associated factors in children and adolescents aged 7 to 14-years-old who were enrolled between the second and ninth years of elementary school in public and private schools.

The sample size estimation was performed based on information from the School Census of 2017 (34,318 students and 82 schools); the outcome of being overweight, including an obesity prevalence of 39% [21,22]; and an acceptable margin of error of 3.5 percentage points, 95% confidence interval and design effect of 1.8. The required sample size was doubled to allow comparisons between previous surveys and to enable subgroup analysis [23], with the addition of 10% for possible losses and refusals. The sample size was estimated at 2891 children and adolescents.

The sampling procedure was performed by conglomerates and had primary sampling units in the schools in the city, which were divided into strata according to the administrative region and type of school. In each stratum, the sample units were randomly selected, totaling 30 schools (19 public and 11 private). The selection of classes was carried out by a systematic random sampling process based on the list of available classes (second to ninth). All students from the selected classes were invited to participate (*n* = 6118) [24].

Schoolchildren who attended the collection day and whose parents/legal guardians signed the Free and Informed Consent Form were included. Anthropometric measurements (weight and height) and dietary intake data were collected from 1671 schoolchildren. Data were excluded for children who had no dietary data (*n* = 188), children who reported implausible dietary data (*n* = 87) (consumption of less than three food items per day or consumption of a number of items greater than the mean + 3 standard deviations, assuming a Poisson distribution for food frequency reports) [25] and for children with missing sleep data (*n* = 63). The final sample included 1333 schoolchildren aged 7–14 years (Appendix A).

This study was conducted in accordance with the guidelines of the code of Ethics of the World Medical Association (Declaration of Helsinki) and approved by the Human Research Ethics Committee of the Federal University of Santa Catarina (UFSC, protocol number 7539718.1.0000.0121).

### 2.2. Assessment of Sleep Data

Sleep data were assessed using a questionnaire with questions adapted from the School Sleep Habits Survey developed and validated by the Bradley Hospital/Brown University in 1994 [26]. The adapted questionnaire was administered to parents or guardians. Regarding sleep timing, the following questions were used: (a) “What time does the child/adolescent usually go to bed on school days?” (b) “What time does the child/adolescent usually wake up on school days?” (c) “What time does the child usually go to bed on weekends (non-schooldays)?” (d) “What time does the child usually wake up on weekends (non-school days)?” Answers to these questions were requested in hours and minutes (local time) [27]. Sleep onset latency and time awake during the night were not assessed.

Total sleep duration was calculated as the difference between bed and wake-up times plus the number of hours slept during the day (nap). Weekday and weekend sleep duration was also calculated. A total sleep duration was calculated as follows: [5 × weekday sleep (h) + 2 × weekend sleep (h)]/7. Bedtimes and wake-up times were described in local time (hh:mm) according to the parents’ answers.

The midpoint of sleep on free days (MSF) was calculated as the halfway point between bedtime and wake-up time [bedtime (local time) + night sleep duration/2] on free days [28]. The hypothesis is that bedtime and wake-up time on free days (without school or work) is highly influenced by the circadian clock [29]. In this study, we considered non-school days (Saturday and Sunday) as free days to calculate the MSF. We also performed the correction of the MSF as proposed by Roenneberg et al. (2004) [30] to compensate for sleep debt accumulated over the schooldays, obtaining the MSF sleep-corrected MSFsc = MSF − 0.5 × (SDF − (5 × SDW + 2 × SDF)/7), where SDF is sleep duration on non-schooldays and SDW is sleep duration on schooldays. For example, a child who goes to bed at 0:00 a.m. and wakes up at 11:00 a.m. on non-schooldays and goes to bed at 21:00 p.m. and wakes up at 7:00 a.m. on schooldays yields a MSF = (0:00 + (11.00)/2) = 5.5 h—or, in local time, 5:30—and a MSFsc = 5.5 − 0.5 × 11.00 − (5 × 10.00 + 2 × 11.00)/7 = 5.14—or, in local time, 5:08. The dimension of MSFsc is not a score but a representation of local time and was transformed into tertiles. In the first tertile, there were those with more morning behaviors, and this group was denominated as “Early”. The second tertile was named “Intermediate”, and the third was named “Late”, composed of schoolchildren with evening preference [31,32].

### 2.3. Socioeconomic Data and Anthropometric Measurements

The type of school (public or private) and school shift (morning or afternoon) were obtained through a list provided by the school administration. Maternal education was classified into three categories (0–8, 9–11 and >12 years) and used as a proxy for socioeconomic status.

Weight and height measurements were performed at the schools by trained researchers following standard techniques [33] and taken from lightly dressed barefoot children. Researchers were trained according to the protocol proposed by the International Society for the Advancement of Kinanthropometry (ISAK) [34]. Weight was measured with a portable digital scale (Marte, model LS200P, 200 kg maximum capacity, 50 g precision). A portable stadiometer (AlturExata, 1 mm precision) was used for height. The body mass index (BMI) was calculated as weight (kg) divided by height squared (m). Age- and sex-specific BMI z-scores were calculated according to World Health Organization criteria for children and adolescents aged 5–19 years [35]. Weight status was categorized as non-overweight (underweight and normal weight, BMI z-score < +1) or overweight, including obesity (BMI z-score for age ≥ +1).

### 2.4. Assessment of Meal and Snack Consumption, Physical Activity and Screen Use

Data on meal and snack consumption, frequency of physical activity and screen use were obtained using the Food Intake and Physical Activity of Schoolchildren (Web-CAAFE) questionnaire, a validated, web-based, self-report tool for use in a school setting. The Web-CAAFE was developed and validated for use with children [36,37] and adolescents [38], considering the cognitive development of 7–10 year-olds [37]. Usability tests showed a child’s capacity to understand and respond to Web-CAAFE [37]. Concerning food consumption, a reproducibility test showed moderate-to-high values of intraclass correlation coefficients [39]. Validity tests of food intake section, using direct observation at school meals as a reference method, showed 43% matches, 29% intrusions and 28% omissions [36].

Children and adolescents were previously instructed by a trained researcher on how to complete the questionnaire with the aid of two banners (140 cm × 105 cm), one with all 31 images of food items and the other with 32 physical and sedentary activities that would be shown on the Web-CAAFE. The researcher explained the concept of each meal and snack and the time of day at which they were consumed, as well as remembering to report the food items eaten from the previous day (yesterday) [36]. 

Web-CAAFE consists of three sections: registration (name, sex, age, weight, height, date of birth and school shift), food intake and physical and sedentary activities. The food intake section is a previous-day recall divided into the consumption of three meals and three snacks, ordered chronologically and presented sequentially on the screen without allocating a specific time in hours: breakfast, mid-morning snack, lunch, mid-afternoon snack, dinner and evening snack. For the food intake section, a robot-like avatar explained the concept of each meal or snack. For example, a child who studies in the morning shift and is answering the Web-CAAFE questionnaire will see and hear an avatar on the screen explaining: ‘Breakfast is the first meal of the day after waking up. Click on the foods you ate for breakfast yesterday’. ‘The morning snack is what you ate after breakfast and before lunch’. ‘This is the snack you usually make at school’. ‘Click on the foods you ate for morning snack yesterday’. These sentences are repeated for each eating occasion to help schoolchildren identify meals and snacks. At the end of each eating event, the avatar explains ‘Remember, if you didn’t eat anything, click on the “nothing” button’. Children who did not consume any food item available in the list of 31 food icons or consumed only water were classified as meal/snack skippers.

Physical activity and screen use were described in detail by Costa et al. (2013) and assessed by three periods of the day (morning, afternoon and night). Daily frequency of physical activity was categorized into tertiles: zero to two times (first tertile), three to four times (second tertile), and five times or more (third tertile). Daily frequency of screen use (watching television, using a computer, using a smartphone/tablet and playing video games) was categorized into never, once a day, twice a day, and more than three times a day [27].

Each child and adolescent answered the instrument once. Web-CAAFE does not provide quantification of the amount of food consumed or the time of eating event, but the six daily eating occasions are ordered chronologically. Also, screen time and physical activity duration are not assessed by the tool, although these activities were presented in three periods of the day. The questionnaire was applied in a school computer room in the presence of trained researchers who assisted the respondents when needed. Data were collected on morning and afternoon shifts and on different days of the week to reflect meal and snack consumption on school days (Monday to Thursday) and a weekend day (Sunday).

### 2.5. Statistical Analysis

Sample characteristics are described as absolute and relative frequencies and 95% confidence intervals (95% CI) for categorical variables. All continuous sleep variables presented non-normal distribution (analyzed by the Shapiro–Wilk test). To investigate the midpoint of sleep differences according to sleep continuous variables, the Kruskal–Wallis test was applied. The Mann–Whitney test was performed to verify median differences in sleep variables by weight status and the frequency of meals and snack consumption. 

The association of the midpoint of sleep (MSFsc) (principal exposure variable) with weight status, meal and snack consumption (outcome variables) was tested using multivariate logistic regression. The intermediate category of midpoint of sleep was considered as the reference. Model 1 considered weight status as the outcome and was adjusted for the following exposure variables: gender, age group (7–10 or 11–14 years), daily frequency of screen use (never, once a day, twice a day and more than three times a day), daily frequency of physical activity (zero to two times (first tertile), three to four times (second tertile) and five times or more (third tertile)), type of school (public/private), maternal education (0–8, 9–11 or ≥12 years of schooling) and total sleep duration (hours). In model 2, meal and snack consumption was the outcome adjusted for variables from model 1, weight status (overweight, including obesity, or non-overweight) and day of food intake report (weekday or weekend). The final models did not include the school shift due to multicollinearity with sleep duration, as previously reported [27]. Due to the proximity of breakfast, mid-morning snack and evening snack with sleep time, the results will be focused on these meals. The analysis performed for lunch, afternoon snack and dinner were not associated with the midpoint of sleep and, therefore, will be shown as a Appendix A (see Appendix A).

Stata version 14.0 (StataCorp LLC, College Station, TX, USA) was used for analysis, and *p* < 0.05 was considered statistically significant. All analyses were adjusted for the survey design effect (using svyset command in Stata).

## 3. Results

The total sample consisted of 1333 children aged 7–14 years. Being overweight (including obesity) was observed in one-third of the sample and was more prevalent in boys than girls. Lunch and dinner were consumed by more than 90% of the children, and a mid-afternoon snack and breakfast were consumed by around 80% of the children, whereas a mid-morning snack (58.7%) and evening snack (54.8%) were consumed by around 50% of the children (Table 1).

Bivariate analysis showed that gender was associated with weight status; type of school; frequency of screen use; mid-morning, mid-afternoon and evening snack consumption; sleep duration and wake-up time. There was a higher proportion of boys who were overweight, including obesity (*p* < 0.001), and girls consumed more mid-morning (*p* = 0.005), mid-afternoon (*p* < 0.001) and evening (*p* = 0.001) snacks.

Students who were overweight or obese had later bedtimes on weekdays (median: 22:00, p25; 75 = 21:30; 23:00) and weekends (median: 23:00 p25; 75 = 22:30; 0:00) than non-overweight students (median: 22:00, p25; 75 = 21:30; 22:30 and median: 23:00, p25; 75 = 22:30; 0:00), respectively. The means were 22:17; 23:18 and 22:09; 23:11, respectively (Table 2).

Breakfast consumers had higher sleep duration in total and on weekdays, an earlier bedtime on weekdays and weekends, an earlier wake-up time on weekends and, consequently, lower MSFsc than breakfast skippers (Table 3).

The differences in bedtime, wake-up time and sleep duration across the three categories of the midpoint of sleep were statistically significant except for sleep duration on weekdays (see Appendix A). Table 4 shows the odds ratio of the associations between the midpoint of sleep categories and weight status. Belonging to the Early group of the midpoint of sleep was associated with lower odds of being overweight (including obesity) (OR: 0.83; 95% CI 0.69; 0.99) after covariate adjustments.

Regarding meal and snack consumption, the Early group had higher odds of mid-morning snack consumption (OR: 1.95; 95% CI 1.56; 2.44), and the Late group had lower odds of mid-morning snack consumption (OR: 0.67; 95% CI 0.55; 0.80) than the Intermediate group (Table 5).

For the evening snack, the students in the Early group had lower odds of consuming this snack (OR: 0.75; 95% CI 0.59; 0.94) compared to the Intermediate group. We also observed that overweight or obese children were less likely to consume breakfast (OR: 0.65; 95% CI 0.47 0.90) and mid-morning snacks (OR: 0.75; 95% CI 0.64; 0.87) than non-overweight children (Table 5). No differences were found in lunch, mid-afternoon snacks and dinner (see Appendix A).

## 4. Discussion

In this present study, we observed an inverse association between being overweight (including obesity) and the Early midpoint of sleep. These cross-sectional results do not allow cause and effect association; however, one possible explanation is that children in the Early group tended to have healthier eating patterns and more physical activities, which may impact weight status [19]. It is also possible that they have more adequate sleeping and waking behavior; therefore, there is no entrainment of circadian rhythms that affect metabolism and weight regulation [40]. For example, if a child stays up late at night, exposed to artificial lights, it causes the entrainment of the circadian cycle, which can affect weight control. Light (and darkness) is the predominant zeitgeber (entraining stimulus) for the circadian clocks [1].

However, we did not observe an association between being overweight and the Late group, in accordance with the results of other studies [19,41]. On the other hand, previous studies reported that later chronotypes were related to higher BMI scores [18,20]. A longitudinal study found an association between the midpoint of sleep and Fat Mass Index, particularly in adolescents between 12 and 15 years of age [15]. 

Children and adolescents with overweight had later bedtimes on school nights and weekend nights than non-overweight students, similar to other studies [10,20]. One possible explanation is that the students who go to bed later may be eating more energy-dense foods that are convenient for late-night eating, as reported by previous studies [17,42,43,44]. Furthermore, Golley et al. (2013) [14] reported that going to bed late/rising late was predictive of a higher BMI and a lower diet quality in children and adolescents. Another study investigated the impact of sleep improvement on food choices in adolescents with later bedtimes. After the intervention, they observed that earlier bedtimes were positively associated with healthy food intake for breakfast [45].

Our study observes no differences in the midpoint of sleep categories of breakfast consumers, but we found they tended to have earlier bedtimes than breakfast skippers. Similar to this finding, other studies reported that schoolchildren who had later bedtimes were more likely to skip breakfast [16,46].). The result related to breakfast is in agreement with a systematic review that showed that children and adolescents who skipped breakfast had an obesity risk of 43% greater than breakfast consumers in cross-sectional studies (no significant link was found in cohort studies) [47]. 

To date, no studies were found in relation to mid-morning and evening snacks and their association with being overweight. We also observed that girls were more likely to consume an evening snack; however, another study found no gender differences in meal skipping [48]. 

In this study, we identified an association between the Early and Late midpoint of sleep and mid-morning and evening snack consumption. The mid-morning associations could be related to the time interval between eating events. A child/adolescent in the Early group wakes up earlier and has an intermediate mid-morning or pre-lunch snack, while a child/adolescent in the Late group who wakes up later and has breakfast does not necessarily have an intermediate snack because there are not many hours for lunch. 

Similarly, a negative association was observed between evening snack consumption and children who had the Early chronotype. One explanation is because they go to bed earlier, and there is not enough time to have a snack after dinner, while children in the Late group supposedly go to bed later and may feel hungry after dinner and before bedtime. Although chronotype was not measured, a study found a positive association between later bedtime and more calories and fat intake after dinner [13]. These hypotheses are supported by differences also found in wake-up time, bedtime and between consumers and skippers of these snacks.

In this present study, food eaten on mid-morning and evening snack was not assessed. Nevertheless, a study performed in public schools in the same city of this study identified an “unhealthy” evening snack pattern composed of a pizza/hamburger/hot dog, chips, sodas, cake and fruit juice consumed by children [49]. 

The use of the midpoint of sleep, bedtime and wake-up time, not only sleep duration, are a strength of this study. It has been shown that sleep habits, not only sleep duration, play an important role related to obesogenic behaviors in children [10]. The decision to use the corrected midpoint of sleep (MSFsc) proposed by Roenneberg et al. (2004) [30] to associate being overweight with meal and snack consumption was based on the concept of better representing the sleep timing and chronotype. This considers sleep timing on both school days and free days and provides an average of the midpoint of sleep over the whole week. Some studies collected sleep data from the night before the date of dietary data collection, aiming to capture the sleep information closest to the same time frame dietary data, but the authors described it as a limitation, as one night of sleep data may not be representative of usual sleep [50,51]. In addition, our study included a sample composed of students from public and private schools, a proxy for socioeconomic position. However, we found no association with this variable, although there are important socioeconomic differences between these groups that may affect sleep timing [52]. The analyses were adjusted to account for physical activities [53], screen use [54], sleep duration [9,19], age and gender, which have recognized associations with sleep patterns and chronotype [2].

This study has some limitations, including a cross-sectional nature that cannot fully demonstrate a clear cause–effect conclusion. Furthermore, sleep data were measured subjectively by parents or guardians using the Sleep Habit Survey; however, subjective measures are used in most large epidemiological studies [55]. Furthermore, Wolfson et al. (2003) [56] published a validation study of the Sleep Habit Survey in adolescents. The results provided evidence supporting the validity of self-reported sleep habits using this questionnaire when the goal was to describe group-level sleep patterns of large samples of adolescents. To date, no validation studies of this questionnaire in the Brazilian population have been observed in the literature. Also, the questionnaire does not provide sleep latency, so we did not adjust the MSFsc for this variable. Meal and snack consumption, frequency of screen use and physical activities were self-reported by children and adolescents, and we cannot exclude the possibility of social desirability and difficulties in recalling past events [37]. The Web-CAAFE avatar explained the concept of each meal or snack to help children and adolescents to identify eating occasions. However, the possibility of errors in the allocation of foods in meals and snacks cannot be excluded. Children younger than 10 years were still in the process of developing cognitive abilities to accurately recall diet and physical activity [57,58]. The dietary and physical activity information derived from the children themselves without help from parents or guardians as the questionnaire was designed, considering the cognitive skills and literacy levels of children aged 7–10 years, has been validated in the study population [36,38]. Also, the use of one-day meal and snack consumption data may not reflect the usual frequency of the consumption of meals and snack [59]. However, this method is widely accepted to assess food intake and meal/snack frequency at the population level [60]. Care was taken to perform data collection on the weekend day (Sunday) and weekdays (Monday to Thursday), allowing us to analyze the variation of diet within the sample, minimizing bias. Dietary intake or meal timing were not evaluated or controlled for, limiting our ability to investigate the influence of these characteristics in the midpoint of sleep. 

The findings of this study elucidated important aspects related to sleep timing associated with being overweight and meal and snack consumption in schoolchildren aged 7–14 years in southern Brazil. These findings suggest that interventions promoting a consistent early bedtime may reduce obesity risk by promoting earlier meal timing [13] and improving healthier food intake [45]. Parental control of sleep timing, including a bedtime routine and earlier betimes, could be important in the long run for children’s and adolescents’ health [61,62] and for weight management concurrently and in the transition to adulthood [63].

## 5. Conclusions

This study identified that children and adolescents with an early midpoint of sleep were less likely to be overweight, and the overweight students had later bedtimes. The consumption of mid-morning and evening snacks was associated with the midpoint of sleep. The Early type of schoolchildren were more likely to consume the mid-morning snack, while the Late type of schoolchildren were less likely to consume this snack. Those of the Early type were less likely to consume the evening snack.

These results suggest that sleep timing is relevant to snack consumption and may also be related to a greater risk of the development of overweight andobesity in children and adolescents. The findings of the current study may provide practical implications for designing interventions aimed at the importance of children’s sleep and eating routines to prevent and reduce being overweight and obese. School interventions and health promotion actions could be proposed to promote earlier bedtimes for children and adolescents who usually sleep later, also aiming to encourage the consumption of adequate meals and snacks throughout the day. 

This study points to some avenues for future lines of research. Future studies may verify differences in the quality or patterns of meals and snacks according to chronotypes. Lastly, longitudinal studies are needed to examine the causal relationship between these characteristics.

## Figures and Tables

**Table 1 ijerph-20-06791-t001:** Description of the sample of 7–14-year-old schoolchildren according to gender. Florianopolis, Brazil, 2018/2019.

Characteristics	Total (*n* = 1333)		Female (*n* = 756)		Male (*n* = 577)		*p* ^b^
	*n*	% (95% CI)	*n*	% (95% CI)	*n*	% (95% CI)	
Gender (*n* = 1333)							
Female	756	53.1 (49.2–57.0)	-	-	-	-	
Male	577	46.9 (42.9–50.8)	-	-	-	-	
Age (*n* = 1333)							
7–10 years	782	57.8 (49.7–65.6)	446	58.0 (48.3–67.2)	331	57.4 (50.2–64.2)	0.550
11–14 years	551	42.2 (34.4–50.3)	310	41.9 (32.8–51.8)	246	42.6 (35.8–49.8)	
Weight status ^a^ (*n* = 1316)							
Non-overweight	876	66.2 (63.0–69.4)	530	73.1 (63.8–80.7)	346	58.4 (53.3–63.4)	**<0.001**
Overweight, including obesity	440	33.8 (30.6–37.1)	218	26.9 (19.3–36.2)	222	41.6 (36.6–46.7)	
Type of school (1333)							
Public	783	58.7 (56.1–61.4))	468	61.9 (58.4–65.3)	315	54.6 (50.5–58.6)	**0.007**
Private	550	41.3 (38.6–43.9)	288	38.1 (34.7–41.6)	262	45.4 (41.4–49.5)	
School shift (1328)							
Morning	746	52.4 (44.9–59.8)	405	53.9 (50.3–57.4)	341	59.2 (55.1–63.2)	0.052
Afternoon	582	47.6 (40.2–55.1)	347	46.1 (42.6–49.7)	235	40.8 (36.8–44.9)	
Daily frequency of physical activity (*n* = 1333)							
zero–two times	804	58.6 (53.7–63.3)	434	58.6 (51.1–65.8)	370	58.5 (52.3–64.4)	**0.045**
three–four times	366	27.2 (23.9–30.8)	222	27.8 (23.9–32.0)	144	26.6 (22.4–31.2)	
≥five times	163	14.2 (11.6–17.3)	100	13.6 (9.4–19.4)	63	14.9 (10.4–20.9)	
Daily frequency of screen use (*n* = 1333)							
never	319	21.1 (15.7–27.8)	165	18.1 (11.9–26.6)	154	24.5 (19.3–30.5)	**0.014**
once	332	26.5 (24.2–29.0)	205	29.3 (24.4–34.6)	127	23.4 (19.2–28.1)	
twice	246	17.3 (15.1–19.7)	127	16.9 (15.8–18.2)	119	17.7 (12.8–23.9)	
three times or more	436	35.1 (31.1–39.2)	259	35.6 (32.4–39.0)	177	34.5 (27.6–42.0)	
Maternal education, years of schooling (*n* = 1288)							
0–8	244	9.1 (2.4–28.5)	138	9.8 (2.7–29.7)	106	8.2 (2.2–26.9)	
9–11	386	22.2 (12.4–36.4)	226	25.2 (15.3–38.6)	160	18.9 (9.0–35.3)	0.625
≥12	658	68.7 (43.0–86.5)	365	65.0 (40.5–83.5)	293	72.9 (45.6–89.6)	
Meals and snacks consumed							
Breakfast	1114	83.0 (80.4–85.4)	632	82.1 (76.8–86.4)	482	84.1 (80.4–87.2)	0.976
Mid-morning snack	753	58.7 (53.4–63.8)	452	62.4 (54.2–70.1)	301	54.5 (50.0–59.3)	**0.005**
Lunch	1293	97.8 (96.6–98.6)	730	97.7 (95.8–98.8)	563	97.9 (96.3–98.8)	0.283
Mid-afternoon snack	1115	88.0 (82.2–87.3)	656	88.3 (84.4–91.2)	459	81.2 (76.8–84.9)	**0.000**
Dinner	1218	92.2 (89.3–94.4)	694	91.4 (86.6–94.6)	524	93.2 (90.2–95.2)	0.526
Evening snack	738	54.8 (50.8–58.8)	449	59.7 (54.1–65.1)	289	49.3 (45.6–53.1)	**0.001**
Day of food intake report (*n* = 1333)							
Weekday	1067	87.6 (70.0–95.6)	598	86.7 (67.9–95.2)	469	88.6 (69.6–96.3)	0.323
Weekend	266	12.4 (4.4–30.5)	158	13.3 (4.8–32.1)	108	11.4 (3.7–30.4)	
	Median (p25; p75)		Median (p25; p75)		Median (p25; p75)		*p* ^c^
Sleep duration (h) (*n* = 1333)							
Total	9.79 (9.14;10.54)		9.86 (9.21; 10.64)		9.71 (9.08; 10.43)		**0.008**
Weekday	9.50 (8.33;10.50)		9.67 (8.75; 10.50)		9.50 (9.92; 10.50)		0.417
Weekend	10.5 (9.50;11.00)		10.50 (10.00;11.17)		10.00 (9.50; 11.00)		**<0.001**
Bedtime (local time) (*n* = 1333)							
Weekday	22:00 (21:30; 23:00)		22:00 (21:30; 23:00)		22:00 (21:30; 23:00)		0.502
Weekend	23:00 (22:30; 0:00)		23:00 (22:30; 0:00)		23:00 (22:30; 0:00)		0.738
Wake-up time (local time) (*n* = 1333)							
Weekday	7:00 (6:30; 8:30)		7:00 (6:30; 9.00)		7:00 (6:40; 8:30)		0.824
Weekend	9:30 (8:30; 10:30)		9:35 (9:00; 10:30)		9:00 (8:30; 10:00)		**<0.001**
MSFsc (*n* = 1333)	3:58 (3:15; 4:38)		3:35 (3:14; 4:38)		3:56 (3:17; 4:38)		0.947

95% CI: confidence interval 95%; p25; p75: interquartile range; MSFsc: midpoint of sleep on free days corrected. ^a^ Classified according to WHO (2006) [35]. ^b^ Pearson’s chi-squared test. ^c^ Mann–Whitney test. Bold values denote statistical significance at the *p* < 0.05 level.

**Table 2 ijerph-20-06791-t002:** Median and mean of sleep variables according to weight status. Florianopolis, Brazil, 2018/2019.

Sleep Characteristics	Non-Overweight (*n* = 876)		Overweight, Including Obesity (*n* = 440)		*p* ^a^
	Median (p25; p75)	Mean (SD)	Median (p25; p75)	Mean (SD)	
Sleep duration (h)					
Total	9.79 (9.14; 10.60)	9.89 (1.10)	9.79 (9.14; 10.43)	9.82 (1.04)	0.291
Weekday	9.50 (8.75; 10.50)	9.68 (1.27)	9.50 (8.83; 10.50)	9.60 (1.20)	0.335
Weekend	10.50 (9.58; 11.00)	10.44 (1.27)	10.08 (9.5; 11.00)	10.37 (1.27)	0.290
Bedtime (local time)					
Weekday	22:00 (21:30; 22:30)	**22:09 (1:01)**	22:00 (21:30; 23:00)	**22:17 (1:04)**	**0.021**
Weekend	23:00 (22:30; 0:00)	**23:11 (1:17)**	23:00 (22:30; 0:00)	**23:18 (1:10)**	**0.020**
Wake-up time (local time)					
Weekday	7:00 (6:30; 9:00)	7:38 (1:23)	7:00 (6:30; 8:30)	7:34 (1:22)	0.829
Weekend	9:30 (8:30; 10:20)	9:28 (1:20)	9:30 (8:30; 10:30)	9:32 (1:22)	0.455
MSFsc	3:56 (3:13; 4:37)	4:02 (1:11)	3:59 (3:18; 4:40)	4:05 (1:05)	0.223

SD: standard deviation; MSFsc: midpoint of sleep on free days corrected. ^a^ Mann–Whitney test. Bold values denote statistical significance at the *p* < 0.05 level.

**Table 3 ijerph-20-06791-t003:** Median of sleep variables according to breakfast, mid-morning snack and evening snack. Florianopolis, Brazil, 2018/2019.

Sleep Characteristics	Breakfast			Mid-Morning Snack			Evening Snack		
	No (*n* = 219)	Yes (*n* = 1114)	*p* ^a^	No (*n* = 580)	Yes (*n* = 753)	*p* ^a^	No (*n* = 595)	Yes (*n* = 738)	*p* ^a^
	Median (p25; p75)	Median (p25; p75)		Median (p25; p75)	Median (p25; p75)		Median (p25; p75)	Median (p25; p75)	
Sleep duration (h)									
Total	9.60 (9.00; 10.29)	9.86 (9.17; 10.64)	**0.005**	9.95 (9.29; 10.64)	9.69 (9.04; 10.45)	**0.001**	9.69 (9.07; 10.36)	9.92 (9.21; 10.68)	**0.002**
Weekday	9.33 (8.50; 10.25)	9.67 (8.83; 10.50)	**0.002**	10.00 (9.00; 10.50)	9.50 (8.65; 10.50)	**<0.001**	9.50 (8.65; 10.33)	9.88 (9.00; 10.50)	**0.001**
Weekend	10.15 (9.50; 11.00)	10.50 (9.50;11.00)	0.467	10.50 (10.00; 11.00)	10.15 (9.50; 11.00)	0.364	10.00 (9.50; 11.00)	10.50 (9.50; 11.00)	0.189
Bedtime (local time)									
Weekday	22:15 (22:00; 23:00)	22:00 (21:30; 22:45)	**<0.001**	22:00 (21:55; 23:00)	22:00 (21:30; 22:30)	**<0.001**	22:00 (21:30; 22:30)	22:00 (21:30; 23:00)	0.071
Weekend	23:30 (22:30; 0:00)	23:00 (22:30; 0:00)	**<0.001**	23:00 (22:30; 0:00)	23:00 (22:00; 23:40)	**<0.001**	23:00 (22:30; 0:00)	23:00 (22:30; 0:00)	0.464
Wake-up time (local time)									
Weekday	7:00 (6:30; 8:30)	7:00 (6:30; 8:50)	0.352	7:30 (6:27; 9:00)	7:00 (6:30; 8.00)	**<0.001**	7:00 (6:30; 8:00)	7:05 (6:30; 9:00)	**<0.001**
Weekend	10:00 (8:39; 11:00)	9:30 (8:30; 10:00)	**0.001**	9:40 (9:00; 10:30)	9:00 (8:30; 10.00)	**<0.001**	9:30 (8:30; 10:15)	9:30 (8:30; 10:30)	0.079
MSFsc	4:03 (3:19; 5:00)	3:56 (3:23; 4:35)	**0.010**	4:09 (3:28; 4:51)	3:27 (3:08; 4:30)	**<0.001**	3:55 (3:21; 4:52)	4:00 (3:29; 4:43)	**0.033**

MSFsc: midpoint of sleep on free days corrected. ^a^ Mann–Witney test. Bold values denote statistical significance at the *p* < 0.05 level.

**Table 4 ijerph-20-06791-t004:** Associations between weight status and the midpoint of sleep categories. Florianopolis, Brazil, 2018/2019.

	Weight Status (Overweight Including Obesity)			
	Crude OR (95% CI)	*p*	Adjusted ^a^ OR (95% CI)	*p*
Early	0.77 (0.58; 1.04)	0.084	0.83 (0.69; 0.99)	**0.043**
Intermediate	1	-	1	-
Late	0.75 (0.41; 1.37)	0.309	0.87 (0.49; 1.55)	0.595
Gender				
Male	1	-	1	**-**
Female	0.52 (0.29; 0.93)	0.032	0.51 (0.27; 0.96)	**0.040**
Age				
7–10 years	1	-	1	**-**
11–14 years	0.92 (0.80; 1.05)	0.202	0.86 (0.75; 0.97)	**0.022**
Screen use				
never	1	**-**	1	**-**
once	0.60 (0.43; 0.84)	**0.008**	0.63 (0.43; 0.91)	**0.020**
twice	0.73 (0.51; 1.04)	0.076	0.77 (0.54; 1.10)	0.127
three times or more	0.86 (0.59; 1.25)	0.389	0.92 (0.65; 1.29)	0.574
Physical activity				
zero–two times	1	-	1	-
three–four times	0.96 (0.57; 1.64)	0.876	0.96 (0.53; 1.75)	0.884
≥five times	0.84 (0.41; 1.75)	0.606	0.81 (0.34; 1.93)	0.597
Type of school				
Public	1	-	1	-
Private	0.90 (0.72; 1.12)	0.302	0.91 (0.70; 1.18)	0.424
Maternal education, years of schooling				
0–8	1	-	1	-
9–11	1.58 (0.90; 2.45)	0.106	1.59 (0.99; 2.51)	0.051
≥12	1.12 (0.73; 1.66)	0.624	1.13 (0.60; 2.11)	0.691
Total sleep duration (h)	0.92 (0.83; 1.02)	0.097	0.90 (0.79; 1.03)	0.108

OR: odds ratio; CI 95%: confidence interval 95%. ^a^ Adjusted by gender, age, screen use, physical activity, type of school, maternal education and total sleep duration. Bold values denote statistical significance at the *p* < 0.05 level.

**Table 5 ijerph-20-06791-t005:** Associations between breakfast, mid-morning snack, evening snack consumption and the midpoint of sleep categories. Florianopolis, Brazil, 2018/2019.

	Breakfast				Mid-Morning Snack				Evening Snack			
	Crude OR (95% CI)	*p*	Adjusted ^a^ OR (95% CI)	*p*	Crude OR (95% CI)	*p*	Adjusted ^a^ OR (95% CI)	*p*	Crude OR (95% CI)	*p*	Adjusted ^a^ OR (95% CI)	*p*
Early	0.98 (0.44; 2.20)	0.966	0.97 (0.43; 2.21)	0.936	1.82 (1.34; 2.49)	**0.002**	1.95 (1.56; 2.44)	**<0.001**	0.83 (0.64; 1.08)	0.138	0.75 (0.59; 0.94)	**0.019**
Intermediate	1	-	1	-	1	-	1	-	1	-	1	-
Late	0.63 (0.31; 1.27)	0.167	0.54 (0.29; 1.02)	0.057	0.66 (0.52; 0.82)	**0.002**	0.67 (0.55; 0.80)	**0.001**	1.34 (0.96; 1.87)	0.078	1.30 (0.99; 1.71)	0.057
Gender												
Male	1	-	1	-	1	-	1	**-**	1	**-**	1	**-**
Female	0.87 (0.54; 1.39)	0.514	0.77 (0.51; 1.16)	0.190	1.38 (0.99; 1.94)	0.058	1.48 (1.01; 2.17)	**0.045**	1.52 (1.21; 1.91)	**0.003**	1.45 (1.21; 1.73)	**0.001**
Age												
7–10 years	1	-	1	-	1	-	1	-	1	**-**	1	**-**
11–14 years	0.73 (0.55; 0.96)	0.029	0.78 (0.44; 1.39)	0.350	1.23 (0.62; 2.46)	0.513	1.14 (0.60; 2.18)	0.660	0.64 (0.45; 0;91)	**0.018**	0.63 (0.46; 0.85)	**0.007**
Screen use												
Never	1	-	1	-	1	-	1	-	1	-	1	-
once	1.05 (0.67; 1.63)	0.821	1.23 (0.67; 2.26)	0.454	0.99 (0.76; 1.29)	0.964	0.95 (0.66; 1.37)	0.763	1.21 (0.79; 1.86)	0.346	1.29 (0.86; 1.95)	0.193
twice	0.72 (0.36; 1.45)	0.318	0.73 (0.39; 1.40)	0.308	0.62 (0.40; 0.97)	**0.038**	0.61 (0.39; 0.96)	**0.037**	0.89 (0.52; 1.54)	0.655	0.94 (0.68; 1.31)	0.704
three times or more	1.01 (0.57; 1.80)	0.973	1.08 (0.51; 2.29)	0.812	1.06 (0.77; 1.45)	0.693	0.92 (0.63; 1.36)	0.654	1.35 (0.64; 2.86)	0.386	1.54 (0.69; 3.47)	0.256
Physical activity												
zero–two times	1	-	1	-	1	**-**	1	-	1	**-**	1	**-**
three–four times	1.53 (0.72; 3.23)	0.231	1.48 (0.74; 2.96)	0.238	1.51 (1.05; 2.17)	**0.030**	1.46 (0.96; 2.22)	0.071	1.42 (1.06; 1.89)	**0.023**	1.55 (1.12; 2.15)	**0.014**
>five times	2.64 (1.95; 3.57)	<0.001	2.33 (1.72; 3.16)	**<0.001**	2.28 (1.08; 4.80)	**0.033**	2.00 (0.80; 5.03)	0.122	3.04 (1.62; 5.72)	**0.003**	2.95 (1.55; 5.65)	**0.004**
Type of school												
Public	1	-	1	-	1	-	1	-	1	-	1	-
Private	1.18 (0.80; 1.73)	0.362	1.08 (0.68; 1.73)	0.710	1.45 (0.95; 2.21)	0.075	1.15 (0.60; 2.20)	0.638	0.83 (0.61; 1.12)	0.193	0.78 (0.52; 1.18)	0.211
Maternal education, years of schooling												
0–8	1	-	1	-	1	-	1	-	1	-	1	-
9–11	0.81 (0.51; 1.30)	0.343	0.85 (0.58; 1.25)	0.362	1.06 (0.57; 1.95)	0.847	1.01 (0.50; 2.02)	0.987	1.07 (0.71; 1.62)	0.713	1.22 (0.80; 1.86)	0.306
≥12	0.98 (0.61; 1.59)	0.943	0.90 (0.53; 1.50)	0.640	1.28 (0.76; 2.14)	0.310	0.94 (0.48; 1.84)	0.830	0.84 (0.68; 1.04)	0.096	1.06 (0.73; 1.52)	0.738
Day of the food intake report												
Weekend	1	-	1	-	1	**-**	1	**-**	1	-	1	-
Weekday	0.90 (0.56; 1.45)	0.638	0.83 (0.51; 1.35)	0.415	2.07 (1.19; 3.61)	**0.016**	1.98 (1.08; 3.62)	**0.031**	0.86 (0.53; 1.41)	0.517	0.87 (0.50; 1.51)	0.587
Weight status												
Non-over-weight	1	-	1	**-**	1	**-**	1	**-**	1	-	1	-
overweight	0.66 (0.42; 1.04)	0.068	0.65 (0.47; 0.90)	**0.015**	0.72 (0.63; 0.83)	**<0.001**	0.75 (0.64; 0.87)	**0.002**	0.67 (0.41; 1.10)	0.101	0.71 (0.38; 1.33)	0.248
Total sleep duration (h)	1.16 (1.02; 1.31)	0.026	1.19 (1.03; 1.38)	**0.023**	0.75 (0.59; 0.97)	**0.034**	0.84 (0.71; 0.99)	**0.039**	1.21 (1.09; 1.36)	**0.004**	1.08 (1.00; 1.16)	**0.040**

OR: odds ratio; CI: confidence interval. ^a^ Adjusted by gender, age, screen use, physical activity, type of school, maternal education, day of food intake report, weight status and total sleep duration. Bold values denote statistical significance at the *p* < 0.05 level.

## Data Availability

The data that support the findings of this study are available from the corresponding author upon reasonable request.

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
