# Peer review of "Association between Sleep Timing, Being Overweight and Meal and Snack Consumption in Children and Adolescents in Southern Brazil"

_ijerph, 2023, doi:10.3390/ijerph20186791_

Round 1

Reviewer 1 Report

While I find this article interesting, it is necessary for the authors to adhere these minor revisions in order for this article to be published:

Title

-          As a recommendation, it is suggested to the authors to slightly modify the title of the article to make it more understandable. A personal suggestion would be the following: “Association between sleep schedule and overweight and snack consumption in children and adolescents in southern Brazil.”

Abstract

-          Authors should divide the abstract into different subheadings.

Materials and Methods

-          Normality tests need to be included

Discussion

-          It is necessary to include practical implications of the study as well as future lines of research.

Author Response

September 1, 2023

Thank you very much for taking the time to review this manuscript. Please find the detailed responses below and the corresponding revisions/corrections highlighted in the re-submitted files.

Reviewer 1                                                                                                                      

While I find this article interesting, it is necessary for the authors to adhere these minor revisions in order for this article to be published:

Title

-          As a recommendation, it is suggested to the authors to slightly modify the title of the article to make it more understandable. A personal suggestion would be the following: “Association between sleep schedule and overweight and snack consumption in children and adolescents in southern Brazil.”                                                                                   

Response: Thank you for the suggestion. We have modified our title. However, we found more appropriate for our study to keep the term "sleep timing" in the title. Similar studies that used Midsleep point (MSFsc) as a marker of the circadian phase or chronotype use the term "sleep timing".

Thivel, D.; Isacco, L.; Aucouturier, J.; Pereira, B.; Lazaar, N.; Ratel, S.; Doré, E.; Duché, P. Bedtime and Sleep Timing but Not Sleep Duration Are Associated with Eating Habits in Primary School Children. J. Dev. Behav. Pediatr. 2015, 36 (3), 158–165. https://doi.org/10.1097/DBP.0000000000000131.

Skjåkødegård, H. F.; Danielsen, Y. S.; Frisk, B.; Hystad, S. W.; Roelants, M.; Pallesen, S.; Conlon, R. P. K.; Wilfley, D. E.; Juliusson, P. B. Beyond Sleep Duration: Sleep Timing as a Risk Factor for Childhood Obesity. Pediatr. Obes. 2021, 16 (1). https://doi.org/10.1111/IJPO.12698

Longo-Silva G, Pedrosa AKP, de Oliveira PMB, da Silva JR, de Menezes RCE, Marinho P de M, et al. Beyond sleep duration: Sleep timing is associated with BMI among Brazilian adults. Sleep Med X [Internet]. 2023 Dec 15 [cited 2023 Aug 31];6. Available from: https://pubmed.ncbi.nlm.nih.gov/37554371/

Abstract

-          Authors should divide the abstract into different subheadings.       

Response: We checked the abstract and it is structured but without subheadings according to the instructions to authors provided by IJERPH.

Materials and Methods

-          Normality tests need to be included                                                             Response: We included in the manuscript that data normality was verified using the Shapiro-Wilk test (page 5, line 204).

Discussion

-          It is necessary to include practical implications of the study as well as future lines of research.                                                                                                                       Response: We included in the Conclusion section one paragraph about practical implications of our study and future lines of research (page 13, lines 442-453).

Reviewer 2 Report

Dear Authors,

This study investigated the relationship between sleep timing and overweight and snack consumption in children and adolescents in southern Brazil, suggesting the need for appropriate adjustment of sleep timing to prevent overweight caused by late-night snack consumption in children and adolescents. The study, aiming to form healthy lifestyle habits in children and adolescents, is interesting, and the statistical analysis method according to the topic is well designed with a large sample.

I recommend a few revisions to this paper for the reader's understanding.

1.     In this study, Saturday and Sunday were considered as non-school days, but it is hard to understand that the data from Web-CAAFE reflected meal and snack consumption on school days (Monday to Thursday) and weekends (Sunday). If the reflection dates are different for each data, it is recommended to clearly state the reasons and situations for this.

2.     I recommend the correct notation for the abbreviation displays.

3.     I recommend accurately displaying information such as the company, region, etc., about the tools used for statistical analysis.

4.     The standard of OR (odds ratio) for each item is missing in the data table, making it a bit difficult to understand the data. It is recommended to present the standard for each item.

5.     I recommend separately marking the p-values where significant results appeared in the data table to increase understanding.

6.     In data table 2, the average value of local bedtime is not shown and only appears in the description. It may be difficult to recognize the difference as the same value is displayed in the table, so it is recommended to display the average value in the table as well.

7.     Although there are precedents showing that bedtime is delayed by screen use, this study did not show significant results with the increase in screen use in all groups of breakfast, mid-morning snack, and evening snack. If you are going to suggest increased screen use as a basis for late bedtime, it is recommended to fully explain in relation to the results in this study or explain through additional correlation analysis. This should also be considered in the interpretation of the results where physical activity increased significantly with more than 5 times in the breakfast and evening snack groups.

8.     I recommend getting corrections on paragraph divisions and number format usage.

Minor editing of English language required

Author Response

September 1, 2023

Thank you very much for taking the time to review this manuscript. Please find the detailed responses below and the corresponding revisions/corrections highlighted in the re-submitted files.

Reviewer  2

  1. In this study, Saturday and Sunday were considered as non-school days, but it is hard to understand that the data from Web-CAAFE reflected meal and snack consumption on school days (Monday to Thursday) and weekends (Sunday). If the reflection dates are different for each data, it is recommended to clearly state the reasons and situations for this.

Response: Thank you for the relevant questions regarding our study. We will provide some reasons to try to clarify some decisions we made in our study.

The meal and snack consumption data were obtained on different days according to the day on data collection was performed (the Web-CAAFE provide data about meals and snacks consumed on the previous day). Because of this, some schoolchildren reported consumption on a weekday (87,6%) and others on a weekend (12,4%). This procedure allowed analyzing the variation of diet within the sample, minimizing bias. We included it in the Discussion section (page 12, lines 421-423).

The sleep habits questions provided data on usual sleep rather than on a specific day. Therefore, it allowed us to calculate the corrected midpoint of sleep (MSFsc) (obtained by MSFsc = MSF - 0.5*(SDF - (5*SDW+ 2*SDF)/7) where SDF is sleep duration on non-schooldays and SDW is sleep duration on schooldays), considering not only weekend but also weekday sleep timing. We described it in Methods (page 3, lines 117-128).

The research group opted to use the corrected midpoint of sleep (MSFsc) proposed by Roenneberg et al., (2007) because this criteria may be more representative of sleep timing and represent the schoolchildren’s chronotype. It considers sleep timing on both school days and free days and provides an average of midpoint of sleep over the whole week.  We described it in Discussion (page 12, lines 385-392).

Finally, to control for possible variations in meal and snack consumption between weekdays and weekends, we adjusted the regression analysis to the day of food intake report, as described in Methods (page 5, lines 218).

  1. I recommend the correct notation for the abbreviation displays.

Response: We corrected the abbreviations.

  1. I recommend accurately displaying information such as the company, region, etc., about the tools used for statistical analysis.

Response: We have included details about the software used in the statistical analysis of the study (page 5, line 224).

  1. The standard of OR (odds ratio) for each item is missing in the data table, making it a bit difficult to understand the data. It is recommended to present the standard for each item.

Response: We included the reference category in Tables 4 and 5 and S2.

  1. I recommend separately marking the p-values where significant results appeared in the data table to increase understanding.

Response: We put the significant p-values ​​in bold in the tables.

  1. In data table 2, the average value of local bedtime is not shown and only appears in the description. It may be difficult to recognize the difference as the same value is displayed in the table, so it is recommended to display the average value in the table as well.

Response: We have included the means of sleep variables in Table 2.

  1. Although there are precedents showing that bedtime is delayed by screen use, this study did not show significant results with the increase in screen use in all groups of breakfast, mid-morning snack, and evening snack. If you are going to suggest increased screen use as a basis for late bedtime, it is recommended to fully explain in relation to the results in this study or explain through additional correlation analysis. This should also be considered in the interpretation of the results where physical activity increased significantly with more than 5 times in the breakfast and evening snack groups.

Response: We agree with your statement regarding screen use. As we did not observe a statistically significant association between chronotype and screen use, we removed this part of the discussion.

  1. I recommend getting corrections on paragraph divisions and number format usage.

Response: We corrected some errors that we identified in the local time in Tables 3 and S1 and revised the paragraph divisions.

Reviewer 3 Report

Dear Authors,

First of all, I congratulate you on the work done and the excellent result. I would also like to note the high relevance of the research, the difficult population group for research (the study of children's opinions requires a careful selection of tools), the detailed design of the research and the well-analyzed results. The article made a good impression on me and I actually have no recommendations. At the same time, I would like to leave a small wish regarding the methods. It would also be interesting to learn from the article which schools (urban, rural) were involved in the research. I also think it is worth writing more about the psychometric / validation points of the School Sleep Habits Survey (is this method validated for the Brazilian population?) I hope these questions will be useful for you and help you improve your article.

Kind regards,

Author Response

September 1, 2023

Thank you very much for taking the time to review this manuscript. Please find the detailed responses below and the corresponding revisions/corrections highlighted in the re-submitted files.

Reviewer 3

Dear Authors,

First of all, I congratulate you on the work done and the excellent result. I would also like to note the high relevance of the research, the difficult population group for research (the study of children's opinions requires a careful selection of tools), the detailed design of the research and the well-analyzed results. The article made a good impression on me and I actually have no recommendations. At the same time, I would like to leave a small wish regarding the methods. It would also be interesting to learn from the article which schools (urban, rural) were involved in the research. I also think it is worth writing more about the psychometric / validation points of the School Sleep Habits Survey (is this method validated for the Brazilian population?) I hope these questions will be useful for you and help you improve your article.                         

Response: Thank you for the comments. This cross-sectional study was conducted as a part of a school-based surveillance named Study on the Prevalence of Obesity in Children and Adolescents in Florianópolis, southern Brazil (EPOCA survey), enrolled in public and private schools in the urban area. We included this in the Methods section (page 2, line 70).

In relation to validation of the School Sleep Habits Survey, we included a phrase in the Discussion section (page 12, lines 402-406).

Reviewer 4 Report

The authors proposed an interesting manuscript aimed to explore relationships between  sleep timing, meal and snack consumption, and weight status in which they present an profile approach of healthier eating patterns among children and adolescents.

This is a relevant topic, but some issues/concerns were faced during the reading of the manuscript and some ideas must be clarify and strengthen in order to have a more solid manuscript:

One of the biggest concerns is that it seems that everyone goes to sleep at the same time from the reports of the parents, but some bias has not been suggested in this regard, why was a more adequate record of the time not made? Or how this could be validated to report.

Also, even though they recognized some limitations it is important to highlight them or explain the relevance of the results despite them, specially on the assessment of snacking, as they mentioned there is a high risk that it has not been correctly evaluated and classified. What measures did they use to ensure that the children understood the message of the avatar?  Especially in the little ones who do not have an adequate management of temporality

On the other hand, the authors highlighted that the midpoint of sleep, bedtime, wake time, not only sleep duration are a strength of this study, but they don’t mention anything regarding the time it takes to fall asleep, that is, a child can "go to bed" early but fall asleep late, even without visual or auditory stimuli. This information this information would be very valuable

 The presentation of the results is clear and easy to read and follow, Just in Table 1. The authors could improve the information providing the statistical test to compare female and males, as it seem to be a confusing variable.

In discussion, there is a lack of information regarding the findings; it seems that those who do not sleep enough are more likely to skip or skip meals. How the authors explain this, is there any kind of compensation? This could be related to the amount of food consumed throughout the day, data that is not presented –or even assessed- . The number of meals is important or energy intake throughout each meal? As well as quality of snacks or meals.

Author Response

September 1, 2023

Thank you very much for taking the time to review this manuscript. Please find the detailed responses below and the corresponding revisions/corrections highlighted in the re-submitted files.

Reviewer 4

The authors proposed an interesting manuscript aimed to explore relationships between  sleep timing, meal and snack consumption, and weight status in which they present an profile approach of healthier eating patterns among children and adolescents. This is a relevant topic, but some issues/concerns were faced during the reading of the manuscript and some ideas must be clarify and strengthen in order to have a more solid manuscript: One of the biggest concerns is that it seems that everyone goes to sleep at the same time from the reports of the parents, but some bias has not been suggested in this regard, why was a more adequate record of the time not made? Or how this could be validated to report.                                                    

Response: We agree with the reviewer and, as another reviewer suggested, we also include the average bedtime in table 2.                                                             We would like to point out that the children's parents were asked to indicate sleep and wake times in hours and minute. Therefore, our questionnaire accepted any time with units in h and min (hh:mm), such as 10:00 p.m. or 10:10 p.m. for example. We included it in Methods (page 3, lines 110, 115).

Also, even though they recognized some limitations it is important to highlight them or explain the relevance of the results despite them, specially on the assessment of snacking, as they mentioned there is a high risk that it has not been correctly evaluated and classified. What measures did they use to ensure that the children understood the message of the avatar?  Especially in the little ones who do not have an adequate management of temporality

Response: On the day of data collection, there was a previous explanation to the children on how to correct respond to the web-CAAFE with the help of illustrated banners. It was explained what each meal was and what foods would appear on the screen so the children could make their own selections.

More details about the questionnaire were included in Methods: “Children and adolescents were previously instructed by a trained researcher on how to complete the questionnaire with the aid of two banners (140cmx105cm), one with all 31 images of food items and the other with 32 physical and sedentary activities that would be shown on the web-CAAFE. The researcher explains the concept of each meals and snacks and the time of day at which is consumed, as well to remembering to report the food items eaten from the previous day (yesterday) [36]”. (page 4, lines 162-167)

The questionnaire includes six eating events ordered chronologically and presented sequentially on the screen (breakfast, mid-morning snack, lunch, mid-afternoon snack, dinner and evening snack) without allocating a specific time in hours. In addition, while the children answer, a robot-like avatar guides children responding to the questionnaire and explains the concept of each meal or snack. For example, a child who studies in the morning shift and is answering the Web-CAAFE questionnaire will see and hear an avatar on the screen explaining: ‘Breakfast is the first meal of the day after waking up. Click on the foods you ate for breakfast yesterday’. ‘The morning snack is what you ate after breakfast and before lunch’. ‘This is the snack you usually make at school’; ‘Click on the foods you ate for morning snack yesterday’. These sentences are repeated for each eating occasion to help schoolchildren identify meals and snacks. At the end of each eating event, the avatar explains ‘Remember, if you didn't eat anything, click on the “nothing” button’.(page 4, line 174-181)                                               

We would like to highlighted that after developing the questionnaire, studies were conducted to test usability, reproducibility, and validity. We provide more details about the validation study in Methods: The Web-CAAFE was developed and validated for use with children [36,37] and adolescents [38], considering the cognitive development of 7-10 year-olds [37]. Usability tests showed child capacity to understand and respond to Web-CAAFE [37]. Concerning food consumption, a reproducibility test showed moderate-to-high values of intraclass correlation coefficients [39]. Validity tests of food intake section, using direct observation at school meals as reference method, showed 43% matches, 29% intrusions and 28% omissions [36]. (page 4, lines 159-161)                                                       

On the other hand, the authors highlighted that the midpoint of sleep, bedtime, wake time, not only sleep duration are a strength of this study, but they don’t mention anything regarding the time it takes to fall asleep, that is, a child can "go to bed" early but fall asleep late, even without visual or auditory stimuli. This information this information would be very valuable

Response: We agreed that it would be valuable information. The questionnaire we used did not allow us to collect sleep latency data, so we did not use the midpoint adjusted for sleep latency. We included this in the limitations of the study (page 12, line 407) and methods section (page 3, line 111).

The presentation of the results is clear and easy to read and follow, Just in Table 1. The authors could improve the information providing the statistical test to compare female and males, as it seem to be a confusing variable.                                                                                        

Response: We included a statistical test to compare females and males in Table 1.

In discussion, there is a lack of information regarding the findings; it seems that those who do not sleep enough are more likely to skip or skip meals. How the authors explain this, is there any kind of compensation? This could be related to the amount of food consumed throughout the day, data that is not presented –or even assessed- . The number of meals is important or energy intake throughout each meal? As well as quality of snacks or meals.         

Response: Thank you for your valuable questions on this topic. However, in this study, it was not possible to test hypotheses related to dietary/energy intake because the WebCAAFE was not designed to provide quantification of the amount of food consumed, therefore does not allow an estimate of total energy intake. We have included this in the methods (page 4, line 192).

We included in the conclusion section some suggestions for further research on this topic (page 13, lines 450-453).                             

In this regard, our research group is working on other studies that aim to elucidate associations between sleep variables and meal patterns using latent class analysis We believe with this study we will be able to verify how sleep variables impact the foods that are consumed by children in the different meals of the day.